# Optical functionalization of human Class A orphan G-protein-coupled receptors

Maurizio Morri[1], Inmaculada Sanchez-Romero[1], Alexandra-Madelaine Tichy[1], Stephanie Kainrath[1], Elliot J. Gerrard[2], Priscila P. Hirschfeld[1], Jan Schwarz[1] & Harald Janovjak[1]

G-protein-coupled receptors (GPCRs) form the largest receptor family, relay environmental stimuli to changes in cell behavior and represent prime drug targets. Many GPCRs are classified as orphan receptors because of the limited knowledge on their ligands and coupling to cellular signaling machineries. Here, we engineer a library of 63 chimeric receptors that contain the signaling domains of human orphan and understudied GPCRs functionally linked to the light-sensing domain of rhodopsin. Upon stimulation with visible light, we identify activation of canonical cell signaling pathways, including cAMP-, $Ca^{2+}$-, MAPK/ERK-, and Rho-dependent pathways, downstream of the engineered receptors. For the human pseudogene *GPR33*, we resurrect a signaling function that supports its hypothesized role as a pathogen entry site. These results demonstrate that substituting unknown chemical activators with a light switch can reveal information about protein function and provide an optically controlled protein library for exploring the physiology and therapeutic potential of understudied GPCRs.

[1] Institute of Science and Technology Austria (IST Austria), Am Campus 1, Klosterneuburg 3400, Austria. [2] Faculty of Biology, Medicine and Health, The University of Manchester, Oxford Rd, Manchester M13 9PL, UK. Correspondence and requests for materials should be addressed to H.J. (email: harald@ist.ac.at)

GPCRs sense diverse extracellular signals and are the targets of ~one third of the available prescription drugs[1–4]. Upon activation by ligands, GPCRs undergo conformational changes and trigger intracellular signaling cascades that are G-protein-dependent or -independent and involve secondary messengers, such as cAMP or $Ca^{2+}$. In the human genome, ~350 genes encode non-olfactory GPCRs, and for ~200 of these receptors, one or several natural ligands have been identified[5,6]. In contrast, limited information is available on the ligands and downstream signaling for the remaining putative GPCRs that are commonly classified as orphan GPCRs[7,8]. Orphan receptors in Class A, the largest GPCR class, exhibit wide tissue distribution, with many expressed ubiquitously and some specifically in the central nervous system, tissues involved in metabolism or the immune system. Many orphan GPCRs are conserved across species and have been implicated in a variety of dysfunctions, including cancer, brain disorders and metabolic disorders[9–12].

In light of their importance as regulators of physiology and as potential drug targets, orphan GPCRs have been studied using various approaches in the past decades. These include screening of libraries to identify natural or man-made ligands and loss-of-function or gain-of-function genetic experiments in vitro and in vivo to reveal the receptor function and the physiological roles. Because ligands and signaling functions could not be identified for many orphan GPCRs, non-signaling modes of action, such as modulation of the activity or the cellular localization of other membrane proteins[13–15], have been proposed. Consequently, major remaining questions are whether orphan GPCRs couple to canonical cellular signaling pathways, and what effects these pathways elicit in specific tissues. Conceptually, the functions of orphan receptors, for GPCRs and also other receptor families, remain unknown as with current technologies it is not possible to activate these proteins in real-time and in situ.

We hypothesized that functionalized orphan GPCRs, in which activation by an unknown ligand is replaced with a known stimulus through rational protein engineering, may enable experimental studies that shed light on their function. We found support for this idea of GPCR functionalization in work describing novel GPCRs created using chimeric domain swapping approaches that combine domains of two parent GPCRs[16]. The underlying rationale is that functional elements of GPCRs, such as the domains responsible for ligand sensing and those responsible for cellular pathway activation, can be combined modularly. Specifically, the brace of the N-terminus, three extracellular loops and seven transmembrane domains are typically associated with the sensory domain responsible for ligand binding and receptor activation, whereas the three intracellular loops (ICL; ICL1, ICL2 and ICL3) and C-terminus are typically associated with the signaling domain responsible for specific downstream transmission (Fig. 1a)[17–19]. Thus, chimeric receptors that link the sensory domain of a non-orphan GPCR with the signaling domain of an orphan or understudied GPCR may be capable of activating similar downstream pathways. We further reasoned that the sensory domain of an opsin, i.e., a light-activated GPCR, would be ideal for this purpose. Light is not only an orthogonal and thus highly specific trigger in the context of most cell types, but also provides activation with high spatial precision (e.g., to address selected cells or tissues in an organism) and temporal precision (e.g., to address selected time points during development)[20–23]. Whereas examples of chimeric GPCRs containing elements of vertebrate opsins and Class A or Class C GPCRs exist[24–29], light activation of GPCRs with unknown function has never been demonstrated.

Here, we designed and engineered a library of chimeric receptors that consist of the extracellular and transmembrane elements of rhodopsin and the intracellular elements of 63 human orphan and understudied GPCRs. When expressed heterologously, we detected coupling to canonical cellular signaling pathways including cAMP-, $Ca^{2+}$-, MAPK/ERK-, and Rho-dependent pathways for a subset of these receptors. This receptor library complements the genome-wide experimental and computational resources for GPCR interrogation[30,31], enables novel optogenetic experiments and provides diverse candidate proteins for structure–function studies.

## Results

**Class A GPCR modularity and chimeric protein design.** We established a platform for the design and functional evaluation of large numbers of light-activated chimeric receptors derived from human orphan and understudied Class A GPCRs (Fig. 1b, c). Past studies of chimeric GPCRs typically focused on receptors from one or a few receptor families and applied diverse design methodologies. Therefore, we first systematically investigated if functional modularity is broadly conserved among Class A GPCRs, and whether primary sequences, which represent the information available for all GPCRs, are sufficient to design functional chimeric receptors. To this end, we created and functionally evaluated chimeric receptors of rhodopsin and nine prominent reference GPCRs from five human Class A GPCR families (Fig. 1c, top). In the sequence-based procedure, we computed a multiple protein sequence alignment and identified the intracellular elements of the nine reference GPCRs using those of rhodopsin as a guide[17]. We next engineered the chimeric genes using a high-throughput genetic engineering method (described in detail below) and expressed the proteins in cultured human embryonic kidney 293 (HEK293) cells. The goal of this experiment was to test if cell signaling responses induced by light stimulation (blue and/or green light, maximal intensity ~400 μW $cm^{−2}$) of the chimeric receptors (Supplementary Data 1, 2) recapitulate those of the corresponding reference GPCRs stimulated by agonists (Supplementary Tables 1 and 2). To detect the activation of canonical GPCR signaling pathways, we primarily applied the transcriptional reporters, in which firefly luciferase (FF) is under the control of one of the three response elements (Fig. 2a). The cAMP response element (CRE) is induced by increased intracellular cAMP levels, such as those encountered downstream of $G\alpha_s$-coupled GPCRs[32]. In turn, CRE activity will be reduced upon lowering of prestimulated cAMP levels by $G\alpha_{i/o}$-coupled receptor signaling. The serum response element (SRE) contains binding sites for ternary complex factor (TCF) and serum response factor (SRF), whereas the minimal SRE (SRE.L) contains only the SRF binding site[33–35]. Both elements are induced downstream of $G\alpha_q$- and $G\alpha_{12/13}$-coupled receptors, either by MAPK pathway activation or RhoA activation[33–35]. We choose transcriptional reporters for this platform because these permitted large-scale screens of light-activated receptor libraries (e.g., 63 receptors, see below) for several reasons. The first reason is that transcriptional assays register pathway activation in stable reporter proteins, and thus allow for relatively long (four to 6 h) experiments with continuous light stimulation. As a consequence of long-term illumination, relatively low light intensities (<500 μW $cm^{−2}$) are sufficient for receptor activation. These intensities can be applied to hundreds of samples in multiple 96-well plates simultaneously using light-emitting diodes (LEDs) and are not associated with unspecific photoeffects or phototoxicity. The second and third reasons are that the activity of multiple signal pathways can be detected using a universal luminescent readout, and that a second constitutively expressed luciferase (Renilla luciferase (RE)) can provide normalization for transfection efficiency and cell viability. Using these assays, we found all nine light-activated chimeric receptors to act on the same pathways as

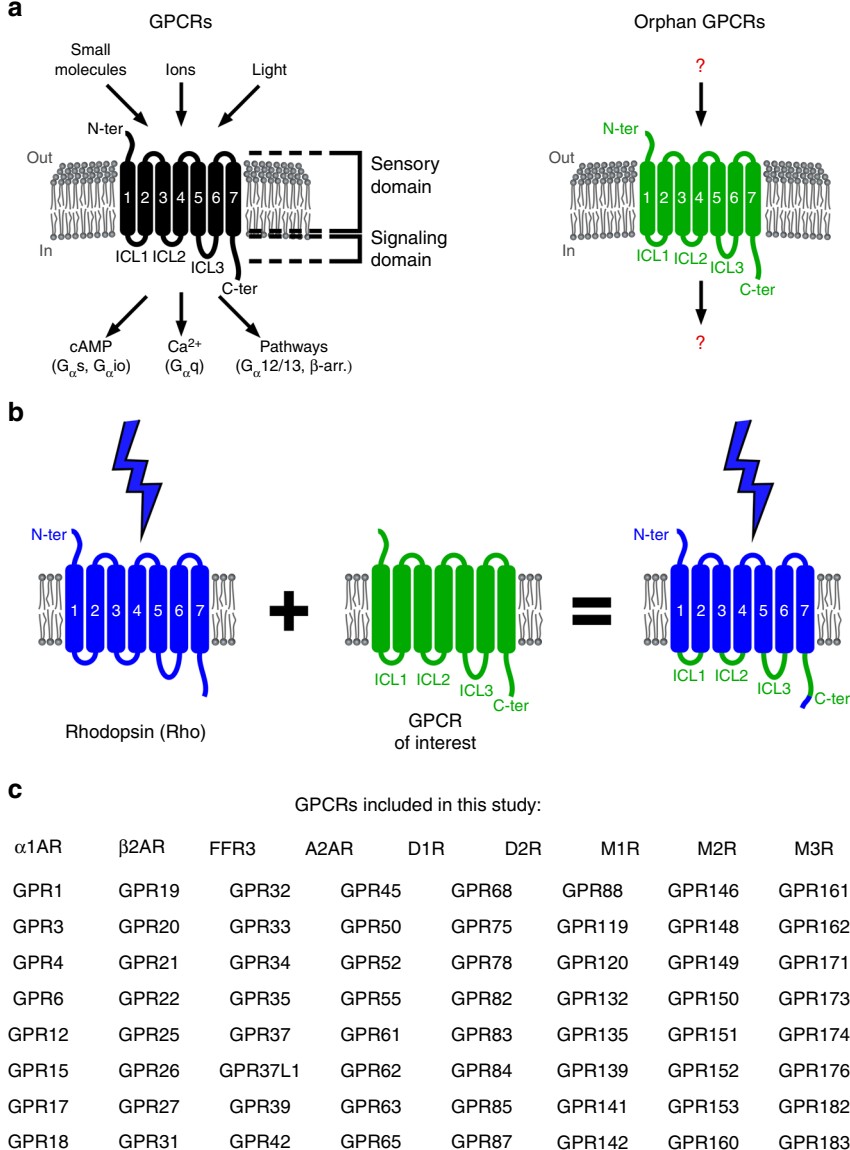

**Fig. 1** Light-activated human orphan and understudied GPCRs. **a** In Class A GPCRs, extracellular and transmembrane domains are responsible for ligand sensing, while intracellular domains couple with downstream signaling pathways. For orphan GPCRs, both the identity of ligands as well as of downstream signals remain unknown. **b** In domain swapping experiments, the intracellular elements of orphan and understudied GPCRs were grafted onto the light-activated GPCR rhodopsin, yielding light-activated, chimeric GPCRs. **c** Reference GPCRs and orphan and understudied GPCRs included in this study

their ligand-activated reference GPCRs (Fig. 2b, c). For instance, stimulation of $G\alpha_s$-coupled or $G\alpha_q$-coupled adrenergic receptors (β2AR or α1AR) with norepinephrine induced CRE and SRE/SRE.L reporters and analogous responses were observed for light-activated Opto-β2AR and -α1AR (Opto- indicates rhodopsin chimeric receptors). Overall, we did not detect false positives (emergence of new functions in chimeric receptors not observed in reference GPCRs) or false negatives (loss of functions observed in reference GPCRs). Furthermore, we found only one receptor, a light-activated dopamine receptor 1 (termed Opto-D1R; Fig. 2c), to exhibit activity in the absence of light. However, even in this case, signaling was further increased by light stimulation. Collectively, these results demonstrate that sequence information is sufficient to design the functional light-activated GPCRs for members of several Class A GPCR families, and that these receptors can be studied in a transcriptional reporter platform.

**Reengineering 63 orphan and understudied GPCRs.** Encouraged by these results, we developed a library of chimeric receptors that contain the intracellular elements of 63 orphan and understudied GPCRs (Fig. 1c, bottom, Supplementary Table 3, Methods). We selected these proteins using a list of Class A orphan GPCRs made available by the International Union of Basic and Clinical Pharmacology (IUPHAR; see Methods)[8]. It is noteworthy that this list contains orphan GPCRs as well as understudied GPCRs, for which ligands have been identified recently. Furthermore, we excluded from our study opsins and proteins that were already assigned to Class A subfamilies with according nomenclature. Preparing the 63 chimeric genes challenged existing genetic engineering strategies because a total of nine alternating, variable-length segments of rhodopsin and the 63 GPCRs (including a far C-terminal epitope of rhodopsin) had to be seamlessly linked (Fig. 1b). We developed a

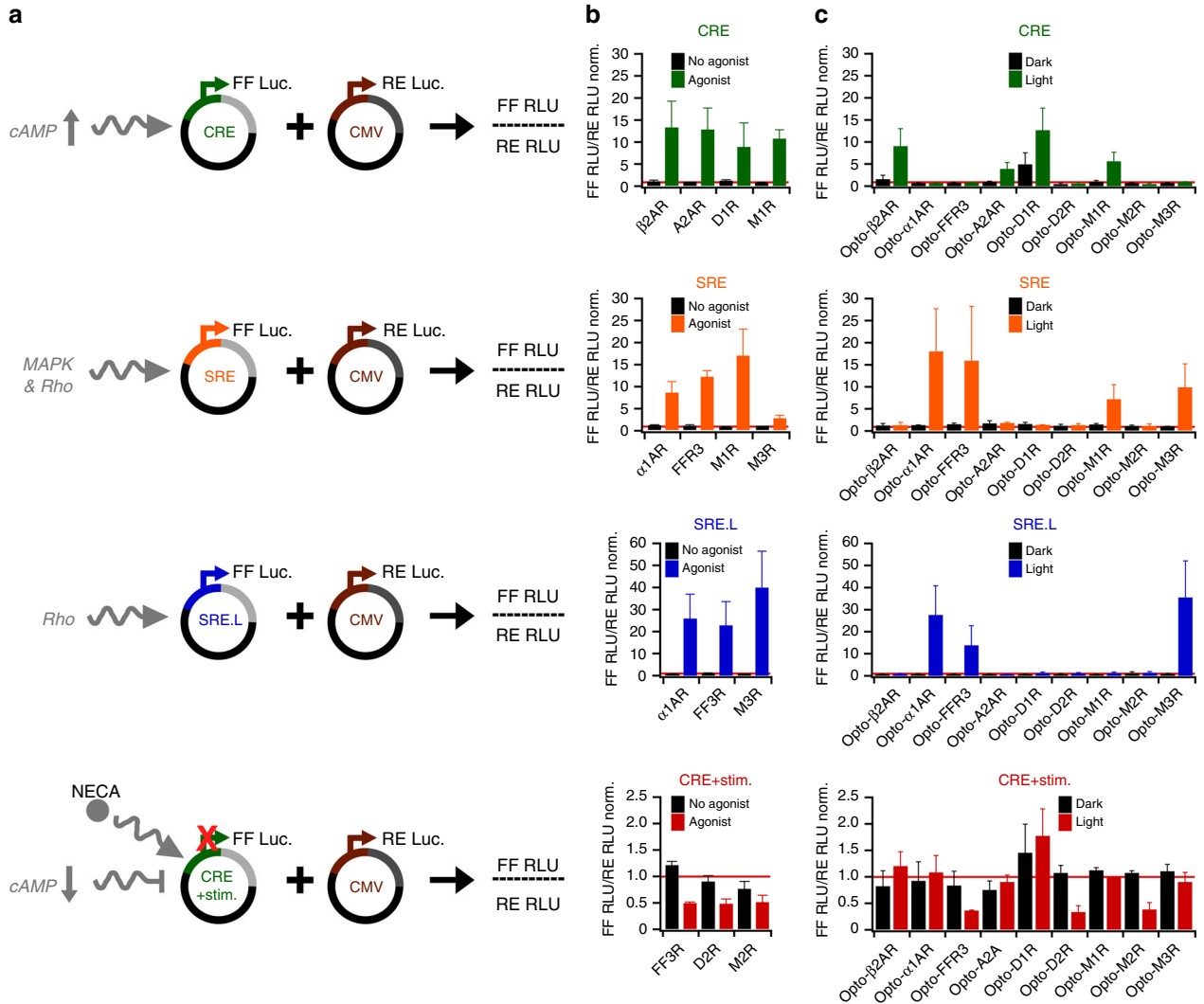

**Fig. 2** Reporter assay and light-activated reference GPCRs. **a** Reporter vectors express firefly luciferase (FF) under the control of signaling-specific enhancers and were co-transfected with *Renilla* luciferase (RE) to normalize for cell numbers/transfection efficiencies. Relative light units (RLU) were detected for both luciferases separately. **b** Reporter activation by reference GPCRs stimulated by agonists (Supplementary Table 2; here, only receptors that stimulated reporters are shown). **c** Reporter activation by the light-activated variants of reference GPCRs. In **b** and **c**, mean values ±s.e.m. ($n = 6$–8, 3–4 independent experiments) are shown. In tests for cAMP reduction, cAMP production was stimulated chemically before light treatment, as described in Methods

multiplexed variant of the Golden Gate protocol[36] to achieve this efficiently. In this method, a modified rhodopsin gene and synthetic gene fragments, corresponding to the intracellular elements of each GPCR, were digested by four bacterial restriction enzymes and ligated by T4 viral ligase in one reaction (Supplementary Figs. 1 and 2, Supplementary Tables 4 and 5). Because clones with correct insertions are selected as the reaction progresses, this method offered high efficiency (>90% clones with correct insertions) and was independent of many reaction parameters (Supplementary Table 6; sequences of the generated genes are given in Supplementary Data 1-4). After genetic engineering, we first investigated the expression of the 63 proteins in the cultured human cells by staining against a vesicular stomatitis virus glycoprotein (VSV-G) epitope in their N-terminus. We observed intensities comparable to that of rhodopsin for 61 receptors (two receptors, Opto-GPR12 and -GPR15, showed lower, but detectable signals) demonstrating efficient chimeric receptor expression (Supplementary Fig. 3).

**Canonical signaling downstream of the engineered receptors.** We next performed a screen of the cellular signaling responses elicited by chimeric receptors. In this screen, we assayed the 63 receptors with the transcriptional reporters in dark and light conditions (blue and/or green light, maximal intensity ~400 µW cm$^{-2}$; each of the 504 receptor-pathway pairs was tested in at least three independent experiments and each experiment consisted of least duplicate wells). We observed specific signaling responses upon light stimulation and generally no activity in the absence of light (Fig. 3, Supplementary Table 7). The most potent induction (threshold: 2.5-fold, see Data Analysis and Statistics) of the CRE reporter was observed for six receptors (Opto-GPR1, -GPR21, -GPR32, -GPR42, -GPR61, and -GPR135; range: 3.40 to 28.6-fold; Fig. 3, green markers), of the SRE reporter for five receptors (Opto-GPR3, -GPR18, -GPR68, -GPR78, and -GPR88; range: 2.70 to 8.20-fold; Fig. 3, orange markers) and of the SRE.L reporter for one receptor (Opto-GPR78; 64.5-fold; Fig. 3, blue markers). We also found potent reduction (threshold: 0.66-fold, see Data Analysis and Statistics) of the CRE reporter for four receptors

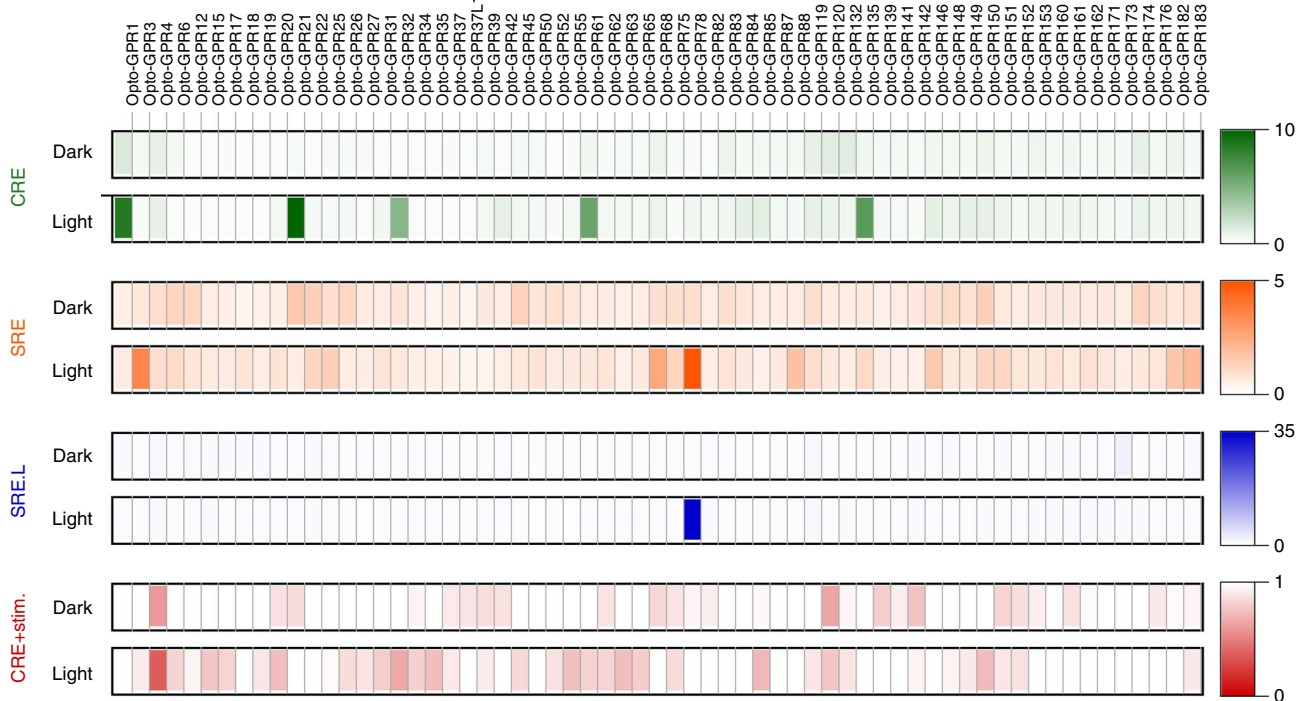

**Fig. 3** Optical functionalization of 63 human GPCRs. Reporter activation by chimeric receptors stimulated with light. Squares are filled with false color representations of mean FF RLU/RE RLU norm. values (n = 6–8, 3–4 independent experiments), as indicated by the scale bars. Also see Supplementary Table 7 for tabulated results

(Opto-GPR4, -GPR55, -GPR63, and -GPR150; range: 0.60 to 0.66-fold; Fig. 3, red markers; cAMP production was stimulated before light treatment, as described in Methods). We validated these findings using secondary messenger assays. For the receptors that induced the CRE reporter, we detected increased intracellular cAMP concentrations using a real-time sensor (Supplementary Fig. 4)[37,38]. For the receptors that induced the SRE/SRE.L reporter, we detected increased intracellular $Ca^{2+}$ concentration using single-cell $Ca^{2+}$ imaging (Supplementary Fig. 5). We did not observe elevated basal cAMP or $Ca^{2+}$ levels in the absence of light, except for cells expressing Opto-GPR68. Furthermore, we confirmed membrane localization of those receptors that elicited these signaling responses using confocal microscopy (Supplementary Fig. 6). Collectively, these results demonstrate specific functions associated with intracellular elements of orphan and understudied GPCRs grafted onto rhodopsin in a functionalized receptor library. For most of these receptors, no signaling functions were reported previously, notable exceptions being proton- or cannabinoid-sensing receptors (GPR4, GPR18, GPR18, GPR55 and GPR68)[39–41].

**Optical resurrection of GPR33.** To demonstrate the further potential of the functionalization approach, we focused on the chemokine-like receptor protein GPR33. GPR33 stands out among the listed receptors because its gene exhibits polymorphic sequence variations including pseudogenization in hominoid (including humans) and rodent species (Fig. 4a)[42,43]. The coincident inactivation of GPR33 in several mammalian species suggests selective pressure on the gene, potentially by a pathogen that employed the protein as an entry site. Experimental support for this hypothesis is missing because the signaling function and internalization behavior of human GPR33 is currently not understood. We addressed this question by resurrecting the ancestral (i.e., not null pseudogene) human *GPR33* allele in a

light-activated protein variant (Opto-GPR33; Fig. 4a, Supplementary Data 3 and 4). We found that light stimulation of Opto-GPR33 reduced CRE reporter activity and induced the SRE/SRE.L reporters (Fig. 4b). This result suggests that the ancestral human GPR33 receptor was capable of activating canonical signaling pathways, including those commonly linked to chemoattractant receptors that are related to GPR33. We furthermore found that light activation of Opto-GPR33 triggered internalization and that the C-terminus was required for this process (Supplementary Figs. 7 and 8). Internalization upon activation may corroborate the hypothesized role of GPR33 as a pathogen interacting and entry factor, and collectively these results illustrate how optical functionalization can be applied to resurrect an ancestral protein function.

**Discussion**

GPCRs critically orchestrate multicellular physiology and are implicated in a plethora of dysfunctions. Despite extensive efforts, ligands for many putative GPCRs, which are classified as orphan receptors, have remained unknown. Understanding the molecular function of orphan GPCRs is essential to gauge their contributions to physiological processes and their potential as drug targets, but this has been hampered by the absence of methods that permit their real-time and in situ activation. In addition, signaling properties of GPCRs, such as preferential coupling to one or several G-protein-dependent pathways, can currently not be predicted based on sequence information due to complex divergent receptor evolution[30]. To overcome these limitations, we generated a functionalized receptor library, in which we grafted the secondary structure elements associated with downstream signaling and trafficking of 63 human Class A GPCRs onto rhodopsin. The receptors we studied were orphan as well as understudied receptors and recently, deorphanized receptors. Functionalization was achieved using an efficient genetic

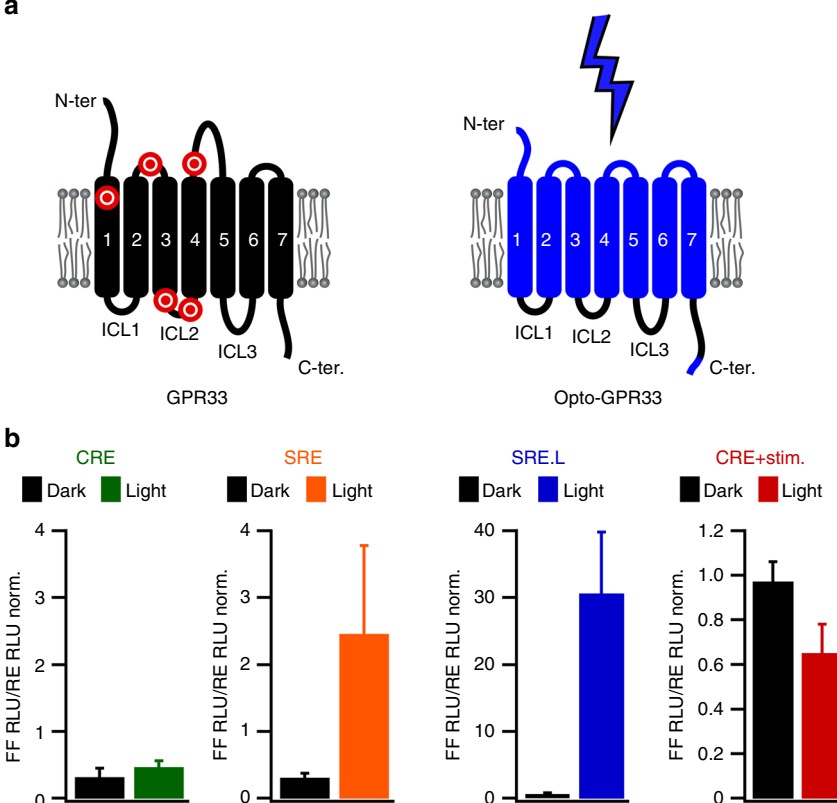

**Fig. 4** Optical resurrection of GPR33. **a** Left: In rodent and hominoid species, the chemokine-like receptor GPR33 is inactivated by stop codons. Red markers: approximate positions of stop codons in the pseudogenized mammalian *GPR33* coding sequences. Right: Light-activated variant of the human ancestral GPR33 receptor. **b** Reporter activation by Opto-GPR33 upon light stimulation. Mean values ± s.e.m. ($n = 6$–8 wells, 3–4 independent experiments) are shown

engineering method that allowed the seamless assembly of multiple gene fragments into complex fusion genes, and that might be applicable also to engineering in other protein families. Our general chimeric receptor approach was inspired by previous studies that employed domain swapping to create GPCRs with desired properties and to study GPCR structure–function relationships[16–19]. However, previous work on chimeric Class A GPCRs typically recruited receptors from one family that are likely structurally compatible as they respond to the same or chemically related ligands and in many cases originate from a common ancestor[30,44]. Furthermore, chimeric receptors that did not show the predicted functionality have also been reported in a small number of cases[45,46]. For our study, it therefore remained to be systematically tested if functional modularity is maintained between multiple Class A GPCR families. To address this point, we compared signaling response elicited by nine reference GPCRs activated by agonists to those of the corresponding chimeric receptors activated by light. We found that optical functionalization recapitulated the coupling specificity in these cases. It was recently suggested that conformational changes during activation of diverse Class A GPCRs converge on conserved contacts on the intracellular side of the receptors (in particular, a conformational switch encompassing residues 3×46, 6×37, and 7×53[47]). As a consequence of this convergence, intracellular interfaces for interaction with downstream partners are likely positioned similarly in chimeric GPCRs and parent GPCRs, providing an explanation for the success of the chimeric protein engineering approach.

We observed potent coupling to downstream pathways for a subset of the studied receptors. The canonical signaling

capabilities identified in our experiments and in future uses of this library may guide drug discovery by identifying drug targets and validating cellular screening assays. Our experiments were focused on G-protein dependent signaling pathways, and it is possible that the chimeric and parent receptors are capable of activating pathways other than those tested here. In particular, receptors may function only in specific cell types, e.g., because of differentially expressed intracellular signaling proteins. The presented receptor library provides a technological basis for uncovering further signaling functions, as the palette of functional assays and cell models is extended.

Highly conserved sequence motifs such as a three residue stretch in transmembrane helix 3 (E/DRY; residues 3×49, 3×50, and 3×51) or the above mentioned contacts near the intracellular receptor surface are critically implicated in conformational changes associated with GPCR activation. Receptors with non-conservative substitutions in these motifs are known to have altered or diminished signaling function[47–49]. We noticed an accumulation of non-conservative and rare substitutions in these motifs in some receptors included in our study, such as in fourteen receptors with at least one non-conservative or unique substitution in both the DRY motif and the three conserved class A contact residues (Supplementary Table 8). These substitutions raise the possibility that these receptors may exhibit non-canonical activation modes and signaling function, and our experimental results corroborate this idea as we did not observe canonical pathway activation for the corresponding chimeric receptors.

The receptor library complements other multireceptor and genome-wide strategies for the interrogation of GPCRs[30,31].

Specifically, light-activated chimeric receptors may enable exquisite control of signaling during development and in a variety of physiological contexts because light can be delivered with spatiotemporal precision in vitro and in vivo. In a complementary strategy, chemically activated chimeric receptors may allow for systemic and sustained regulation of signaling[50]. It is important to note that the strength of pathway activation, which is known to be biased by GPCR ligands, is likely not preserved in engineered chimeric receptors. One additional use of the chimeric receptor library may be to provide templates for structure elucidation building on the past uses of the exceptionally stable rhodopsin core. Finally, the general approach of substituting activation by unknown ligands or binding partners for activation by light may be extended to other protein families because functional modularity and domain swapping are not limited to GPCRs[51,52].

## Methods

**Availability of vectors and genes**. All vectors and genes created in this study are available through Addgene.org and from the authors.

**Reference GPCR expression vectors**. Reference GPCRs (Fig. 1c, top, Supplementary Table 1) were obtained in mammalian expression vectors, except for D1R, D2R, M1R and M2R that were subcloned into pcDNA3.1(−) using polymerase chain reaction (PCR) and NotI, KpnI and HindIII restriction enzymes (oligonucleotide primers 1 to 8, Supplementary Table 9). All genes were verified by DNA sequencing and tested by agonist activation in the same experiments as the chimeric receptors.

**Gene selection and curation**. In April 2011, we retrieved the complete list of 94 expressed Class A GPCRs that were initially classified as orphans from the online database of IUPHAR (http://www.guidetopharmacology.org/GRAC/FamilyDisplayForward?familyId=16). Twenty-nine genes that were either opsins or assigned to GPCR families with corresponding names were not considered for experiments in this study but included in sequence alignments. Protein and mRNA sequences were collected and curated manually where necessary: GPR79 was excluded because its mRNA sequence could not be obtained, and for three proteins the corresponding mRNA sequences were adjusted at single sites to match the deposited protein sequences. Also, the recognition sequences of restriction enzymes used in the cloning steps were removed from the nucleotide sequences using macros written in Igor Pro (WaveMetrics).

**Design of chimeric GPCR sequences**. A multiple sequence alignment generated using Muscle[53], and a series of macros written in Igor Pro were used to design the chimeric receptors. First, protein and nucleotide sequences (including those of rhodopsin) were imported into Igor Pro and assigned unique identifiers. Using macros, protein sequences were then randomized in their order, exported, aligned with Muscle and reimported into Igor Pro. The intracellular elements (four for each receptor: ICL1 to ICL3 and C-terminus) were then identified in the protein alignment using the corresponding elements of rhodopsin[17] as a guide. The corresponding nucleotide sequences were retrieved and modified with overhangs for genetic engineering. Complete C-termini were included. For further validation, the intracellular elements determined using the above methodology were compared to those predicted by membrane helix analysis. Except for one receptor (GPR101), only minor differences between these two methods were found. The chimeric GPR101 sequence is unlikely to result in a functional protein.

**Engineering of chimeric GPCR genes**. We devised a multisite cloning technique, based on TypeIIS restriction enzymes and Golden Gate cloning[36], for generation of the chimeric receptors (Supplementary Fig. 1). Our technique differs from the original Golden Gate protocol, in that we used four enzymes with distinct recognition sequences and one additional digestion step. We first prepared a mammalian expression vector (Supplementary Fig. 2, Supplementary Table 5). This was necessary because common mammalian expression vectors are not suited for cloning with TypeIIS restriction enzymes due to recognition sites in the vector backbones. In our vector, based on the commercial vector pcDNA3.1(−) (LifeTechnologies), six recognition sites for three TypeIIS restriction enzymes (two for BsaI, one for BpiI/BbsI and three for LguI/SapI) were removed using site-directed mutagenesis (Supplementary Fig. 2, Supplementary Table 5, oligonucleotide primers 9 to 20, Supplementary Table 9). Next, a receiver gene was designed and synthesized that contains (i) the extracellular and transmembrane elements of bovine rhodopsin, (ii) sequences for membrane integration and antibody staining, and (iii) TypeIIS recognition sites in opposing orientation (Supplementary Fig. 1, Supplementary Table 4). This gene was inserted

into the new vector using XhoI and EcoRI restriction enzymes. Synthetic gene fragments corresponding to the intracellular elements of reference or orphan and understudied GPCRs (determined as described above) were obtained as oligonucleotides (Integrated DNA Technologies) or synthetic genes (GeneWiz) and inserted into the receiver at a single or two sites in parallel. Reaction mixtures (typical final volume 20 μl) contained vector (100 ng), gene fragments corresponding to the intracellular elements (one or two, not necessarily adjacent; 7.8 ng each), T4 ligase (1 μl), T4 ligase buffer and 5 units of the TypeIIS restriction enzyme required to insert one of the two elements. This mixture was then cycled 11 times between 37 °C (for 2 min) and 16 °C (for 3 min). Next, five units of the TypeIIS restriction enzyme required to insert the other element was added, followed by further cycling (22 cycles). The enzymes were then heat-inactivated (50 °C for 5 min, 80 °C for 10 min) and the vectors were digested with five units of an additional restriction enzyme to remove unmodified receiver genes (AgeI, BlpI, EcoRV or NotI; Supplementary Table 4). The reaction was then transformed into competent E. coli cells, and one to two clonal DNA preparations were obtained (two clonal DNA preparations were sufficient because the success rate of single fragment insertions was ~90%). The procedure was insensitive to the number of cycles in the range from 33 to 60, but improved by the additional digestion step (Supplementary Table 6). All genes were verified by DNA sequencing. A small number of genes that could not be obtained using the cloning technique right away were prepared using gene synthesis (EpochLifeScience). Note that for future transfer or subcloning of the generated genes using PCR, universal subcloning oligonucleotides using AclI, AgeI, AsISI, BlpI, EcoRV, NotI, PacI, PmeI, and SalI are suited, as recognition sites for these enzymes are not found in any of the genes.

**Opto-GPR33 and its variants**. Opto-GPR33 was designed and constructed, as described above. The fluorescent protein mCherry[54] was a kind gift of R.Y. Tsien (University of California San Diego). To obtain Opto-GPR33-mCherry, an AscI restriction site was introduced before the stop codon of Opto-GPR33 using an inverse PCR (oligonucleotide primers 21 and 22, Supplementary Table 9). In this reaction, amplification produced linear double-stranded DNA products with terminal AscI restriction sites. Products were digested with AscI, ligated, and propagated in E. coli cells. mCherry was then inserted after PCR using the AscI restriction enzyme (oligonucleotide primers 23 and 24, Supplementary Table 9). To obtain Opto-GPR33-BT, the biotinylation tag (BT; GLNDIFEAQKIEWHE) was introduced at the protein N-terminus after the epitope with a flexible linker using an inverse PCR and phosphorylated oligonucleotides with split BT overhangs (oligonucleotide primers 25 and 26, Supplementary Table 9). Opto-GPR33-mCherry and Opto-GPR33-BT lacking the GPR33 C-terminus (Opto-GPR33-mCherry-ΔCt and Opto-GPR33-BT-ΔCt) were obtained by inverse PCR using oligonucleotides that bind to the retained sequences and contain ClaI restriction sites (oligonucleotide primers 27 and 28, Supplementary Table 9). All genes were verified by DNA sequencing.

**Cell culture and transfection**. HEK293 cells (American Type Culture Collection; further authenticated by assessing cell morphology and growth rate) were cultured in a mycoplasma-free environment in DMEM supplemented with 10% FBS, 100 U ml$^{-1}$ penicillin and 0.1 mg ml$^{-1}$ streptomycin in a humidified incubator (37 °C, 5% $CO_2$). Cells were lifted with trypsin (LifeTechnologies) and transfected in DMEM supplemented with 5% FBS, either using Lipofectamine 2000 (LifeTechnologies) or a homemade reagent (1 mg ml$^{-1}$ polyethylenimine in $H_2O$; Polysciences). For transfection, vectors in Opti-MEM I (see below for volumes and amounts) and transfection reagents in an equal volume of Opti-MEM I were combined, incubated at room temperature for 5 min, and added to the cells. Medium was replaced with complete or starve medium (see below) after 6 h.

**Transcriptional reporter assays**. FF reporter vectors were obtained from Promega (CRE reporter: pGL4.29, SRE reporter: pGL4.33)[35] or a kind gift of D. Wu (Yale School of Medicine, SRE.L reporter)[34]. We chose transcriptional reporters because they cover pathways with direct relevance for gene regulation by GPCRs[35] and are suited for experiments with light-activated receptors (Main Text). The RE vector was obtained by subcloning the RE gene from phRG-TK into pcDNA3.1(−) using PCR and NheI and XbaI restriction enzymes (oligonucleotide primers 29 and 30, Supplementary Table 9). In pcDNA3.1(−), RE is under constitutive control of the CMV promoter. 50,000 cells/well were seeded in poly-L-ornithine (PLO)-treated (Sigma) 96-well plates (white or black walls, clear bottoms; Greiner Bio-One). 6 h after transfection with 75 ng receptor vector, 75 ng FF vector and 7.5 ng RE vector in 25 μl Opti-MEM I, the medium was changed to starve medium (DMEM supplemented with 0.5% FBS and antibiotics). 24 h after transfection, the cells were incubated with 10 μM 9-cis retinal overnight. On the next day, the medium was changed to $CO_2$-independent medium (LifeTechnologies) supplemented with 0.5% FBS and antibiotics, and the cells were stimulated with agonists (Supplementary Table 2) or light for 6 h. Light stimulation was performed in an incubator (PT2499, ExoTerra) modified with 450 LEDs (300 IP65, SMD3528, 150 IP66, SMD5050, 470 and 530 nm peak wavelength). Cells were stimulated at 37 °C with blue

and green light (~400 µW cm$^{-2}$), blue light only (~280 µW cm$^{-2}$, for SRE reporter), or shielded from light (wrapped in aluminum foil in the incubator). When testing for a reduction of CRE reporter signals, cells were stimulated with 500 nM 5′-(N-ethylcarboxamido)adenosine 3 (NECA) for 5 min before the agonist or light stimulation. Luminescence was developed using a dual luciferase assay (Dual-Glo, Promega) following the manufacturer's instructions for cell lysis, substrate addition and quenching. RLU were determined in a microplate reader (Synergy H1M, BioTek; gain 135, 1 s integration time).

**cAMP mobilization**. A genetically encoded real-time cAMP sensor (GloSensor-22F[37]) was obtained as a synthetic gene in pcDNA3.1(−). A total of 10,000 cells/well were seeded in 96-well plates and transfected with 100 ng receptor vector and 100 ng sensor vector. 6 h after transfection, the medium was changed to assay medium (Leibovitz's L15 medium supplemented with 10% FBS and antibiotics). 24 h after transfection the cells were incubated with 10 µM 9-*cis* retinal overnight. On the next day, cells were incubated with 2 mM beetle luciferin (Promega) that was reconstituted in 10 mM HEPES. Cells were equilibrated in the dark for 15 min at 37 °C, followed by recording RLU before (for 30 min) and after (for 15 min) light stimulation. Light stimulation was performed using a custom, external array of high-intensity LEDs (HLMP-CE35, Broadcom; ~9.6 mW cm$^{-2}$, 505 nm peak wavelength) for 2 to 10 s. Recordings were analyzed using initial ten data points (for baseline) and the maximum value after stimulation (for percentage change; Supplementary Fig. 4).

**Ca$^{2+}$ mobilization**. A total of 50,000 cells/well were seeded on poly-L-lysine (PLL)-treated 12 mm glass coverslips and transfected with 500 ng receptor vector and 50 ng of a mammalian expression vector harboring the red fluorescent protein mKate2[55](a kind gift of D.M. Chudakov, Shemiakin-Ovchinnikov Institute of Bioorganic Chemistry/Evrogen). 6 h after transfection, the medium was changed to complete medium. 24 h after transfection, cells were incubated with 10 µM 9-*cis*-retinal overnight. For imaging on the next day, the coverslips were washed in measurement buffer (5.4 mM KCl, 135 mM NaCl, 1.8 mM CaCl$_2$, 0.9 mM MgCl$_2$, 10 mM HEPES, 10 mM glucose, pH 7.6) and incubated with 5 µM Fura2-AM (Anaspec) for 25 min in the dark. Coverslips were transferred to fresh buffer to allow dye maturation and were imaged on a customized inverted microscope (Olympus IX50; ×20 magnification) equipped with a high-intensity light source (DG-4, Sutter Instruments) and EMCCD camera (Luca, Andor). Using MicroManager[56], excitation wavelengths were alternated between 340, 380 and 470 nm for dye imaging and photoactivation (3 s frame rate, 400 frames). Individual cells were identified in the mKate2 images recorded prior to Ca$^{2+}$ measurements. Using pattern recognition macros written in Igor Pro, Fura2 fluorescence signals for individual cells were analyzed as time courses (baseline: average of initial ten data points, response: maximum value; Supplementary Fig. 5).

**Opto-GPR33 internalization using mCherry**. A total of 80,000 cells were seeded on PLL-treated glass imaging dishes (MatTek Corporation) and transfected with 500 ng receptor vector (either Opto-GPR33-mCherry or Opto-GPR33-mCherry-ΔCt). Images were acquired 24 h after transfection on a digital fluorescence microscope (EVOS-FL, ThermoFisher Scientific, ×20 magnification, excitation and emission wavelengths were 531 ± 40 and 593 ± 40 nm, respectively). Cells were classified (Type A: cells with internalized receptors, Type B: cells without internalized receptors) and counted manually but blinded to the experimental condition. Dishes were stimulated with blue light (~180 µW cm$^{-2}$) in the incubator described above for 30 to 60 min and analyzed before and after light stimulation. Ratios of Type A cell counts and Type B cell counts define the internalization ratios that are given in Supplementary Fig. 8a.

**Opto-GPR33 internalization using biotinylation and labeling**. A total of 30,000 cells/well were transfected in 96-well plates with 100 ng BirA$^{ER}$ vector that encodes an ER-resident biotin ligase (a kind gift of A. Ting, Stanford University) and 100 ng Opto-GPR33-BT, Opto-GPR33-BT-ΔCt or a transmembrane-bound extracellular BT (TMD-BT; a kind gift of O. Thoumine, University of Bordeaux[57]). 6 h after transfection, the medium was replaced and supplemented with 10 µM biotin (Sigma-Aldrich). After 48 h, cells were incubated with 10 µM 9-*cis*-retinal overnight. On the next day, the cells were washed with DPBS and labeled with 10 µg ml$^{-1}$ HyLiteFluor555 (AnaSpec/Kaneka Eurogentec SA) for 10 min at 37 °C. After washing, the cells were placed in the incubator described above either in blue light (~180 µW cm$^{-1}$) or protected from light for 1 h. Cells were fixed with 4% PFA for 10 min and ten images per well were acquired on the digital fluorescence microscope. Cells were classified (Type A: cells with complete receptor internalization, Type B: cells with complete or partial membrane localization) and counted manually, but blinded to the experimental condition. Ratios of Type A cell counts and Type B cell counts are given in Supplementary Fig. 8b.

**Receptor expression using HRP and confocal microscopy**. A total of 20,000 cells/well were seeded in 96-well plates and transfected with 100 to 150 ng receptor vector. After 48 h of transfection, the cells were washed with PBS and fixed with

4% PFA. After fixation, the cells were washed, blocked with 1% BSA in PBS, and incubated with antibodies against the N-terminal VSV-G epitope (for HRP detection: clone P5D4, V5507, Sigma; 1:250 final dilution in blocking buffer; for confocal microscopy: Dylight™ 488 conjugated antibody, 600-441-386, Rockland, 1:500 final dilution in blocking buffer). For the HRP-based detection, cells were washed and developed using a polymer detection system (Ultravision, Empire Genomics). A mixture was prepared by adding 30 µl of DAB in 1 ml of DAB substrate (Sigma) to the cells after two additional washes. The reaction was stopped by adding water. Optical density at 450 nm was measured in the plate reader after one wash and addition of 30 µl of fresh water. For confocal microscopy, cells were washed and covered with mounting medium. Pictures of individual cells were recorded on an inverted confocal microscope (LSM 700, Zeiss, ×63 objective, excitation and emission wavelengths were 488 and 500–700 nm, respectively.). Unlike confocal microscopy, the plate reader-based HRP measurements do not report on receptor membrane localization, as labeling of intracellular epitopes cannot be excluded even under non-permeabilizing conditions.

**Data analysis and statistics**. In transcriptional reporter assays (Fig. 2b, c, Fig. 3, Fig. 4b, Supplementary Table 7), FF RLU were divided by RE RLU yielding [FF RLU / RE RLU], and thereby correcting for variability in cell number and transfection efficiency[58]. For all samples on a given 96-well plate, [FF RLU / RE RLU] ratios were normalized by those of uninduced control wells that contained the same reporter on the same plate (yielding [FF RLU / RE RLU] norm.), thereby allowing comparison of values collected in independent experiments. For each reporter vector, three to five independent experiments were performed for each receptor each in duplicate wells with mean values given. Thresholds for induced or reduced receptor levels of 2.50- and 0.66-fold, respectively, were defined using the data of Fig. 2b that show below/above threshold responses for reference GPCRs. Transcriptional reporters were analyzed separately with no cross-reporter comparison or normalization. For instance, in Fig. 3 a linear color scale was applied to each assay separately, as maximal levels of reporter induction also reflect reporter sensitivity and not necessarily the relative strength of pathway activation.

In antibody staining (Supplementary Fig. 3), the mean optical density was determined at five positions in each well. Mean optical density values were normalized by the mean optical density of rhodopsin transfected cells. For each receptor, three independent experiments were performed.

For cAMP mobilization, five independent experiments were performed in triplicate wells. Baseline and responses were defined as described above and means are given in Supplementary Fig. 4a and b.

For Ca$^{2+}$ mobilization, three to five independent experiments were performed. In each experiment, 15 single cells were analyzed. Baseline and responses were defined as described above and means are given in Supplementary Fig. 5a and b.

For Opto-GPR33-mCh(-ΔCt) internalization, cells from at least ten images per condition and experiment were analyzed as described above. Experiments were performed in duplicate or triplicate dishes and means are given in Supplementary Fig. 8a.

For Opto-GPR33-BT(-ΔCt) internalization, cells from at least 20 images per condition were analyzed as described above. Experiments were performed in triplicate wells and means are given in Supplementary Fig. 8b.

**Data availability**. Data supporting the findings of this manuscript are available from the corresponding author upon reasonable request. All vectors and genes created in this study are available through Addgene.org.

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

## Acknowledgements

We thank K. Kolev and M. Spanova for technical assistance, C. Heidsiek for assistance with imaging, S. Marillonnet for advice on TypeIIS restriction enzymes, K. Singh for advice on receptor engineering, I. Chamma and O. Thoumine for advice on biotinylation tag experiments, P. Traunmüller and T. Adletzberger for electrical and mechanical engineering, and C. McKenzie, K. Groschner, D. Wu, R. Lefkowitz, G. Milligan, R.Y. Tsien, D.M. Chudakov, and A. Ting for cAMP reporter, fluorescent protein, GPCR and BirA genes and vectors. This work was supported by a grant of the European Union Seventh Framework Programme (CIG-303564 to H.J.) and by the graduate program MolecularDrugTargets (Austrian Science Fund (FWF): W1232).

## Author contributions

M.M. designed and performed genetic engineering, transcriptional reporter experiments, and HRP-based detection. I.S.-R. designed and performed confocal microscopy. A.-M.T. designed and performed cAMP measurements. S.K. designed and performed BT internalization experiments. E.G. designed and performed residue conservation analysis. P.P.H. designed and performed $Ca^{2+}$ measurements. J.S. designed and tested the genetic engineering method. H.J. contributed to the design

of all experiments, performed genetic engineering, performed mCh internalization experiments, contributed to residue conservation analysis, conceived and supervised the study, and wrote the manuscript with input from all authors.

## Additional information

**Competing interests:** The authors declare no competing interests.

