## [Peer Review File · Nature Communications]

Editorial Note: This manuscript has been previously reviewed at another journal that is not operating a transparent peer review scheme. This document only contains reviewer comments and rebuttal letters for versions considered at Nature Communications. Mentions of prior referee reports have been redacted.

REVIEWERS' COMMENTS:

Reviewer #1 (Remarks to the Author):

The authors have done a nice job attending to the prior concerns and I think this will be a very useful resource for those who study GPCRs. I have some relatively minor concerns which are mainly editorial in nature.

SPECIFIC CONCERNS:

1. In the original review, I recommended referring to these as understudied and orphan GPCRs and referring to that as oGPCRs rather than orphan GPCRs as in the paper. Although IUPHAR currently (and out of date) considers those GPCRs orphans several certainly are not. Thus, GPR68 and 65 for instance are bona fide pH sensors (Huang et al Nature 2015) and there is essentially no debate about this (although the IUPHAR website is not up-to-date). I would recommend adding that caveat as this is likely true for many of those listed as orphan.
2. There are other genome-wide resources available to interrogate oGPCRs and these should be mentioned in the Introduction and Discussion (Kroeze et al, Nature Structure Biol 2015) and recent excellent reviews on GPCRs (Wacker et al, Cell 2017) might be considered referring to.

Reviewer #2 (Remarks to the Author):

This manuscript has been transferred to this journal along with my original comments. The authors have addressed my previous comments adequately and the manuscript is much improved. It is an impressive body of work and will leave a lasting impression. I am happy to recommend it for publication.

REFEREE #1 (Remarks to the Author):

REFEREE #1: The authors have done a nice job attending to the prior concerns and I think this will be a very useful resource for those who study GPCRs. I have some relatively minor concerns which are mainly editorial in nature.

AUTHORS: We thank the referee for her/his positive final comment.

REFEREE #1: 1. In the original review, I recommended referring to these as understudied and orphan GPCRs and referring to that as oGPCRs rather than orphan GPCRs as in the paper. Although IUPHAR currently (and out of date) considers those GPCRs orphans several certainly are not. Thus, GPR68 and 65 for instance are bona fide pH sensors (Huang et al Nature 2015) and there is essentially no debate about this (although the IUPHAR website is not up-to-date). I would recommend adding that caveat as this is likely true for many of those listed as orphan.

AUTHORS: We thank the referee for this comment that allowed us to significantly improve the manuscript.

We now clearly mention in the revised manuscript that we generated a receptor list with help of the IUPHAR database and that this list not only includes orphans but also and importantly deorphanized/understudied receptors (please refer to pages 2, 4, 8 and 11 of the revised manuscript).

We agree with the reviewer that the abbreviation "oGPCR" was not used adequately, also in light of above realization and our previous revisions. We therefore decided to remove this abbreviation and instead at each specific instance precisely define whether we refer to orphan receptors, understudied receptors or both. We think this greatly improved clarity in the manuscript.

REFEREE #1: 2. There are other genome-wide resources available to interrogate oGPCRs and these should be mentioned in the Introduction and Discussion (Kroeze et al, Nature Structure Biol 2015) and recent excellent reviews on GPCRs (Wacker et al, Cell 2017) might be considered referring to.

AUTHORS: We thank the referee for pointing these studies out; we have cited these articles in the revised manuscript (please refer to e.g. pages 3, 4 and 13 of the revised manuscript).

REFEREE #2 (Remarks to the Author):

Referee #2: This manuscript has been transferred to this journal along with my original comments. The authors have addressed my previous comments adequately and the manuscript is much improved. It is an impressive body of work and will leave a lasting impression. I am happy to recommend it for publication.

AUTHORS: We thank the referee for her/his positive final comment.